# In vitro role of Rad54 in Rad51-ssDNA filament-dependent homology search and synaptic complexes formation

Eliana Moreira Tavares[1], William Douglass Wright[2], Wolf-Dietrich Heyer [2], Eric Le Cam [1] & Pauline Dupaigne [1]

Homologous recombination (HR) uses a homologous template to accurately repair DNA double-strand breaks and stalled replication forks to maintain genome stability. During homology search, Rad51 nucleoprotein filaments probe and interact with dsDNA, forming the synaptic complex that is stabilized on a homologous sequence. Strand intertwining leads to the formation of a displacement-loop (D-loop). In yeast, Rad54 is essential for HR in vivo and required for D-loop formation in vitro, but its exact role remains to be fully elucidated. Using electron microscopy to visualize the DNA-protein complexes, here we find that Rad54 is crucial for Rad51-mediated synaptic complex formation and homology search. The Rad54 −K341R ATPase-deficient mutant protein promotes formation of synaptic complexes but not D-loops and leads to the accumulation of stable heterologous associations, suggesting that the Rad54 ATPase is involved in preventing non-productive intermediates. We propose that Rad51/Rad54 form a functional unit operating in homology search, synaptic complex and D-loop formation.

[1] Genome Maintenance and Molecular Microscopy UMR8126 CNRS, Université Paris-Sud, Université Paris-Saclay, Gustave Roussy, F-94805 Villejuif Cedex, France. [2] Department of Microbiology and Molecular Genetics, University of California, Davis, Davis, CA 95616-8665, USA. Correspondence and requests for materials should be addressed to P.D. (email: pauline.dupaigne@gustaveroussy.fr)

Homologous recombination (HR) is a major pathway for the repair of broken chromosomes using mainly an intact sister chromatid as a template[1,2]. HR is also engaged in the repair of interstrand crosslinks, single-stranded gaps, and in the recovery of stalled and collapsed replication forks[2]. Consequently, defects in HR are associated with genetic instability, chromosomal aberrations, carcinogenesis, and cell death[3].

The RecA/RadA/Rad51 family of proteins conducts the signature reactions of homology search and DNA strand invasion during HR. These recombinase scaffolds are nucleotide cofactor-modulated proteins that form right-handed helical filaments on single-stranded DNA (ssDNA)[4]. Assisted by other proteins, the nucleoprotein filaments perform homology search by probing duplex DNA throughout the genome to find a complementary DNA sequence required as template for DSB repair. The DNA in the ATP-bound filament is non-uniformly stretched to 150% of B-form DNA, with three-nucleotide segments with a near normal B-form distance between bases followed by a long untwisted stretch between triplets[5]. This remarkable feature likely facilitates base-flipping of triplet units and provides the structural basis for homology search, facilitating homology recognition by triplet base increments[6]. The DNA bases and sugars are alternately stacked facilitating the angular displacement of bases[5,7,8]. Since homology search is faster in A–T rich sequences, it has been proposed these bases flip out of the helix more readily to serve as nucleation sites of homology probing[9,10]. Homology search relies on probing tracts of eight-nucleotide microhomology based on the transient interactions between the stretched single-stranded DNA within filament and bases in a locally melted or stretched DNA duplex[9–15].

When homology is found, the interaction between the nucleoprotein filament on ssDNA and the duplex DNA donor results in their incorporation into a three-stranded intermediate, the synaptic complex, also known as a paranemic joint[16–18]. RecA-mediated synaptic complexes in which DNA strand pairing is maintained by the RecA filament are sensitive to deproteinization, whereas heteroduplex DNA, in which the invading ssDNA and its complementary DNA strand in the dsDNA donor are intertwined, is resistant to deproteinization[19,20]. In the latter case, the strand intertwining is a result of DNA strand invasion forming heteroduplex plus a displaced strand, yielding a structure called the Displacement-loop (D-loop). The D-loop is an important HR intermediate since its formation is required to prime DNA synthesis by the 3′ OH of the invading strand[21]. Many studies have contributed to the better understanding of homology search and D-loop dynamics; however, the mechanistic steps leading to Rad51-mediated synaptic complexes are incompletely characterized, especially concerning the involvement of other proteins in addition to Rad51.

Despite the similarity between RecA and eukaryotic Rad51 nucleoprotein filament structure, and their roles in homology search and DNA strand invasion during HR, both proteins display significant differences in their biochemical mechanisms. RecA exerts a much more robust ATPase activity compared to Rad51 showing a 100-fold difference (~20 min$^{-1}$ versus 0.2 min$^{-1}$)[22]. Moreover, RecA can form D-loops, the invasion of a single-stranded DNA into a supercoiled duplex DNA, autonomously, whereas budding yeast Sacharomyces cerevisiae Rad51 requires its partner protein Rad54[23,24]. Human RAD51 can form D-loops on its own, but only in the presence of calcium, and this activity is highly stimulated by human RAD54[25–27]. Rad54 is a dsDNA-specific ATPase[24,28] and molecular motor with dsDNA translocase activity[29], whose ATPase activity is specifically stimulated by Rad51 bound to dsDNA[30–32]. Rad54 has been proposed to act as a heteroduplex pump that drives D-loop formation, simultaneously removing Rad51 as it generates new

heteroduplex DNA[21]. The Rad54 N-terminal domain binds both Rad51 and ssDNA-containing junction DNA, and both stimulate Rad54 ATPase activity[21,33,34]. It has been suggested that Rad54 creates a heteroduplex junction branchpoint and uses motor activity to extend the heteroduplex DNA in the D-loop, and reverse it in some contexts[21]. The potential for a role of Rad54 preceding D-loop during homology search is indicated by the association of Rad54 with the Rad51-ssDNA filament[35,36]. The presence of Rad54 stabilizes the Rad51-ssDNA filament in vitro and in vivo independently of its ATPase activity[37,38]. This may contribute to more effective homology search, and results from Rad51 Chromatin Immunoprecipitation (ChIP) experiments support this conjecture[39]. The mechanism involved remains to be determined, whether more efficient homology search is an indirect consequence of filament stability or whether Rad54 plays an active role in steps leading to synaptic complex formation.

In this study, we investigate the role of Rad54 in homology search and synaptic complex formation. We use electron microscopy (EM) to directly observe and characterize the molecular features of the DNA-protein intermediates generated at different time points during the homologous pairing reaction. Surprisingly, we find that Rad54 is essential not only for the formation of D-loops but also for homology-independent DNA probing and protein-mediated synaptic complex formation. Unlike bacterial RecA, Rad51-ssDNA filaments do not interact autonomously with dsDNA in a duplex capture assay, but do so robustly in the presence of Rad54. This interaction is not dependent on homology in the dsDNA and may represent a probing activity in homology search. We analyze the D-loop architecture and show evidence for Rad51 removal from the synaptic joint (strand invasion site). This process is mediated by Rad54 and differs from the reaction catalyzed by RecA protein. The Rad54K341R ATPase mutant is identified as a separation of function protein as it supports formation of Rad51-mediated synaptic complexes but not D-loops, showing that ATP hydrolysis by Rad54 is not required for this step. Furthermore, reactions with Rad54K341R exhibit higher frequencies of heterologous DNA associations, suggesting that Rad54 limits the accumulation of stable heterologous engagements by the nucleoprotein filament. We integrate theseinsights into a comprehensive model of the concerted activities of Rad51 and Rad54 in the homology search, synaptic complex and D-loop formation steps of HR.

## Results

**Visualization of D-loops using Electron Microscopy (EM).** In this study, we use a DNA substrate composed of 5′ 609 base pairs and 3′ overhang of 831 nucleotides to mimic the structure (5′ junction) and approximate length of a processed DSB in vivo (Fig. 1a). A supercoiled dsDNA is the homologous donor, allowing easy distinction between invading DNA and topologically closed target DNA by EM. We first verify that Rad51 filaments are well formed on the 5′ DNA junction and that the supercoiled donor DNA is well spread on the EM grids (Fig. 1b). The 5′ junction DNA with only RPA presents a similar structure as in Fig. 1c, in which the dsDNA (609 bp) can be measured (average 200 nm) and the ssDNA part (831 nt) is covered with RPA, with undefined structure reflective of the flexible and stochastic binding of RPA to ssDNA. The Rad51 filament covers not only the ssDNA part of the invading 5′ DNA junction but also dsDNA. In some cases protein-free dsDNA can be observed.

Rad51 filaments are formed on the 5′ DNA junction, with further addition of RPA, followed by Rad54 and supercoiled homologous DNA. The moment of Rad54 and dsDNA addition corresponds to time 0 of the time course. DNA-protein intermediates that are formed in the reaction are directly

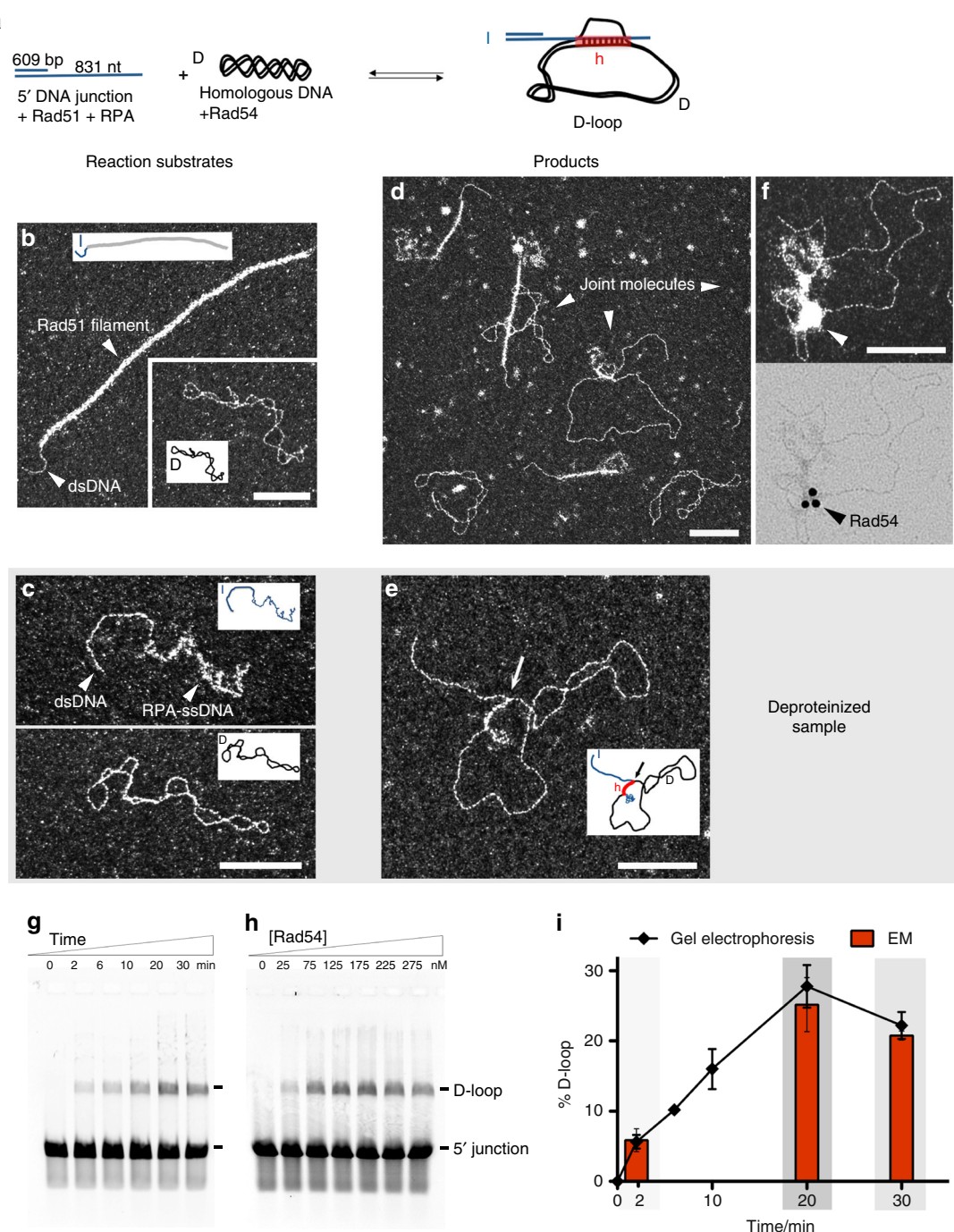

**Fig. 1** D-loop assay by electron microscopy (EM) and by gel electrophoresis. **a** Scheme of the D-loop assay. **b** Representative EM images of the reaction substrates with insets of schematic drawings of the molecules: the Rad51 filament formed on 5′ DNA junction and the homologous supercoiled DNA. **c** Reaction substrates in the deproteinized samples suitable for EM analysis: 5′ DNA junction with 3′ ssDNA part covered with RPA (top), supercoiled homologous DNA donor (middle). **d** EM representative view of the D-loop reaction after 20 min incubation with homologous DNA and Rad54. **e** D-loop observed in the deproteinized sample, with an arrow pointing to the synaptic contact between molecules (bottom). For additional images see Supplementary Fig. 2a. **f** Imunolabeling of Rad54 in the D-loop reaction using rabbit anti-Rad54 primary antibody and immunolabeled anti-rabbit secondary antibody followed with DNA-protein complexes purification. The same joint molecule is imaged using dark-field mode in the top image while using brightfield (bottom image) allowing the detection of the gold beads. Seven 1% of labeled joint molecules where counted ($n = 2$, 100 molecules each). **g** D-loop assay time course after donor DNA addition using 175 nM of Rad54 **h** D-loop assay with Rad54 titration in 20 min reactions. D-loop yield peaks at 175 nM of Rad54, which corresponds to 7 Rad54 proteins per invading or donor dsDNA molecule. **i** Graph of data in **c** (line plot, from electrophoresis gel quantification, $n = 5$) and from the EM quantification (red bars plot, at three time points: 2, 20, and 30 min. $n = 3$, 300 molecules each). Error bars indicate standard deviations. Gray shading corresponds to the three time points chosen for further experiments. In all EM pictures, the white bar represents 100 nm. In **a**, **d**: I invading strand (blue), D donor DNA (black), and h heteroduplex (red). Source data are provided as a Source Data file

visualized using EM revealing a mixture of different types of molecules (Fig. 1d). We clearly distinguish joint molecules resulting of the interaction between the nucleoprotein filament and the duplex DNA donor, coexisting with the reaction substrates (free Rad51 filaments on 5′ junction and supercoiled DNA). To be able to identify the presence of Rad54 in the structure of the protein:DNA complexes, we carried out immunolabeling EM experiments using anti-yeast Rad54 primary antibodies and immunogold secondary antibodies (Fig. 1f, Supplementary Fig. 1). We detect immunogold beads in the contact zone of some joint molecules in reactions with Rad54 (7/100) but none (0/100) in control reactions lacking Rad54 or using a primary antibody directed against another yeast protein (Srs2). These results show that Rad54 can specifically be localized in this region of the complexes. Interestingly, Rad54 is also specifically detected inside some Rad51 filaments (16/100) (Supplementary Fig. 1), consistent with a previous report with human RAD54[36].

The D-loop is defined as a joint molecule product of the incorporation of the invading strand into a homologous dsDNA donor, resulting in the disruption of its original base pairing replaced by a newly formed heteroduplex. The traditional D-loop assay involves deproteinization of the reaction products so that the DNA species can be separated on a gel and quantified. In the D-loop products, the invading strand is intertwined with its complement in the donor, and is stable in the absence of proteins. We developed a deproteinization method suitable for EM observation to compare to the traditional D-loop assay analyzed by gel electrophoresis (Fig. 1g–i) and to gain more information about the D-loop structure. This deproteinization treatment is based on the addition of EDTA to disrupt Rad51 filaments by removal of $Mg^{2+}$/ATP[40]. Raising the temperature to 40 °C for 10 min inactivates Rad54[21]. This allows for EM analysis of D-loop yield and structure (Fig. 1e). It is likely that some RPA still remains bound to ssDNA, but this does not affect the analysis. In controls without Rad54, only isolated 5′ junctions and donor DNA molecules but no D-loops are found at 6, 20, and 30 min, in congruence with the electrophoretic analysis (Fig. 1h). In the EM images, the D-loop is easily detected, as the dsDNA part of the invading 5′ junction molecule is associated with a duplex plasmid. In the presence of Rad54, we count 6% of D-loops at 2 min, reaching a peak of 25% at 20 min with a decline to 21% at 30 min, matching the results obtained by gel electrophoresis (Fig. 1g, i).

In the absence of Rad54, no D-loops are detected by gel electrophoresis (Fig. 1h), consistent with previous observations[24]. When Rad54 is titrated into the D-loop reaction, products reach a peak at 175 nM Rad54. Using this protein concentration, a time course yields 6% D-loops after 2 min, increasing to a peak of 28% at 20 min, with a small decline to 22% at 30 min (Fig. 1i).

EM is ideal to directly observe different populations of DNA and protein:DNA complexes. In the case of the D-loop reaction this affords the opportunity to study the intermediates that precede strand intertwining, specifically protein-mediated pairings such as synaptic complexes where the invading DNA is not intertwined with the donor molecule, these joint molecules being susceptible to deproteinization.

**Rad54 is crucial for the formation of synaptic complexes.** We set out to investigate the molecular structure of the DNA species generated during the homologous pairing reaction by EM observation of DNA-protein complexes (without deproteinization). Reactions lacking Rad54 will not form D-loops, but to our surprise we do not observe Rad51-mediated joint molecules either. Most Rad51 filaments on 5′ junction DNA remain unengaged with dsDNA donor even 20 min after addition of dsDNA. In these reactions lacking Rad54, paired molecules constitute less than 2%, similar to reactions lacking donor homology (Figs. 2a, d and 3b, d, control—no Rad54). The few complexes that are identified have very short contact length (<10 nm), sometimes perpendicular to the dsDNA of the donor and the donor usually remains supercoiled, suggesting these complexes are not mediated by homology recognition (Supplementary Fig. 2). Some supercoiled dsDNA contain short Rad51 filament patches, which lead to its relaxation.

Strikingly, when Rad54 is added to the reaction, Rad51 filaments on 5′ DNA junctions are found paired with donor plasmid molecules, forming synaptic complexes. Rad51-mediated synaptic complexes are the protein-containing joint molecules resulting from the homologous alignment of Rad51-ssDNA with complementary donor sequence. Here, we score protein-mediated pairings of the invading and donor DNA molecules. A fraction of these pairings are likely between heterologous sequences in the DNAs, though a heterologous control suggests this fraction is low in reactions containing wild-type Rad54 (see below). As expected for a kinetic intermediate preceding the D-loop products, synaptic complexes accumulate earlier than D-loops. At 2 min, 19% of Rad51-mediated synaptic complexes are formed, increasing to 20 and 25% at 20 and 30 min (Fig. 2b, d). These observations demonstrate that Rad54 is absolutely required to form Rad51-mediated synaptic complexes. In the absence of Rad54, reactions with both homologous and heterologous dsDNA donors show very low DNA pairing yields of less than 2% (Fig. 3d).

Importantly, these protein-mediated pairings promoted by Rad54 were almost exclusively homology-dependent. In reactions containing heterologous dsDNA, pairing is less than 4% (Fig. 3b, d). Qualitatively, they present distinct features in comparison to the homologously-paired synaptic complexes. These nucleoprotein filament interactions with heterologous DNA are mainly characterized by short contacts (<10 nm) and the dsDNA does not become topologically relaxed (Supplementary Fig. 3). In contrast, homologous pairings have longer contact lengths in synaptic complexes and the donor is proportionately relaxed (see below).

**Rad54 promotes formation of protein-free D-loops.** Interestingly, in the DNA-protein samples with Rad54, we find two main types of joint molecule structures: the Rad51-mediated synaptic complexes, and D-loops. We divide them into these two main categories based on the presence or absence of Rad51 filament in the contact zone (synaptic joint). In synaptic complexes, Rad51 filaments remain present on the contact zone between the 5′ DNA junction and the dsDNA donor (Fig. 2b, Supplementary Fig. 2). D-loops are here scored as the joint molecules in which Rad51 has been displaced from this synaptic zone, where no Rad51 filaments reside on the heteroduplex DNA (Fig. 2c, Supplementary Fig. 2). D-loops show a yield of 2% at 2 min, reaching a peak of 14% at 20 min and decreasing to 12% at 30 min (Fig. 2d). This D-loop yield is lower than determined by gel electrophoresis and EM of deproteinized samples (Fig. 1i). The difference is possibly due to: (i) Rad51 filaments on the synaptic joints being partly disrupted, resulting in an underestimate of D-loops in EM by this measure, (ii) the Rad54 ATPase may allow a subset of formation of heteroduplex within the Rad51 filament (still in a Rad54 ATPase-dependent manner), or (iii) Rad51 left on the ssDNA outside of the heteroduplex region after removal during heteroduplex formation is able to repolymerize back into the synaptic region.

In order to highlight the specific role of Rad54, we also analyze joint molecules architecture formed during D-loop assays with *E.*

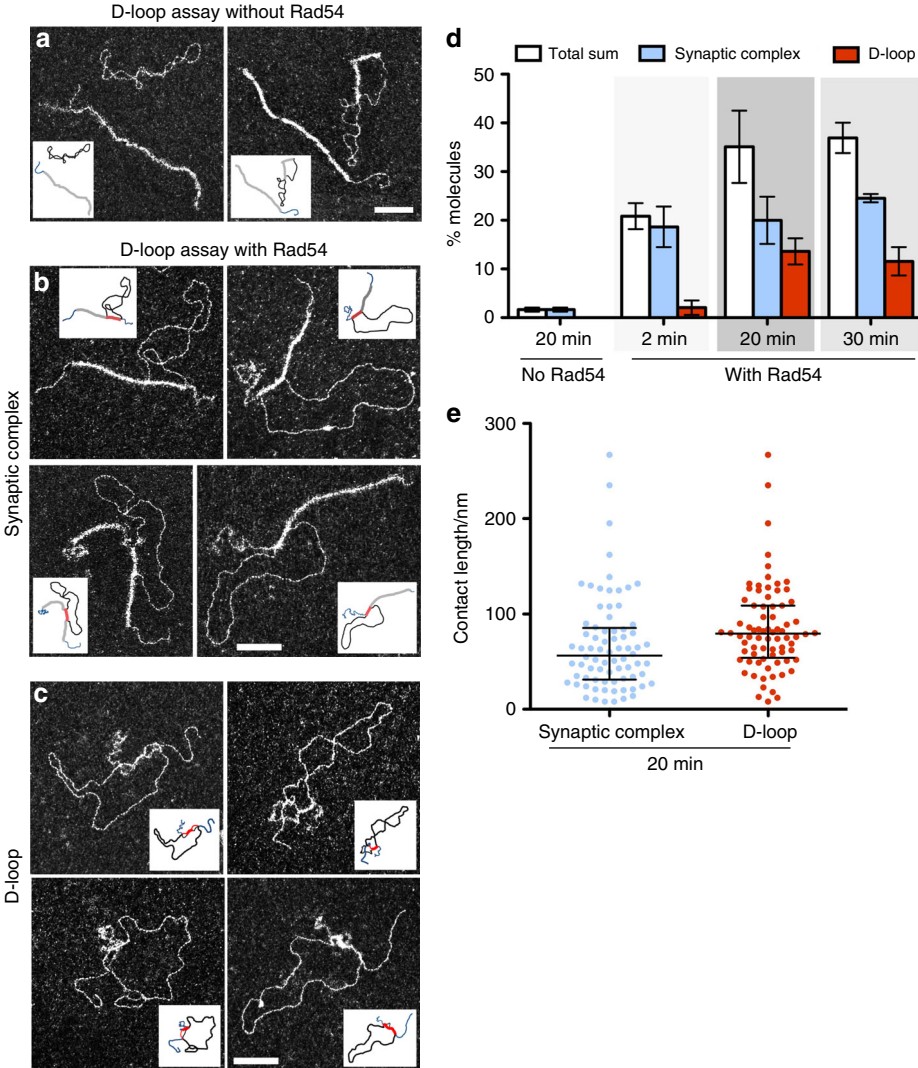

**Fig. 2** Rad54 promotes the formation of synaptic complexes and D-loops as visualized by EM of DNA-protein complexes. **a–c** Representative EM pictures of **a** reactions without Rad54, **b** Rad51-mediated synaptic complexes and D-loops **c** from reaction containing Rad54. In all EM pictures, white bars represent 100 nm. For additional images see Supplementary Fig. 1b, c. **d** Quantification of synaptic complexes and D-loops in DNA-protein samples. Error bars in **d** indicate standard deviation from three independent experiments. The percentage of molecules was determined by the ratio between the population of synaptic complexes or D-loops and the sum of these two categories of molecules plus free 5′ DNA junctions. ($n = 500$ molecules for each experiment). Note that this division into two categories was performed by EM visual analysis therefore rare subjective analysis-associated errors cannot be rolled out. **e** Contact length for synaptic complexes and deproteinized D-loops at 20 min, showing median with interquartile range. Source data are provided as a Source Data file

*coli* RecA recombinase, which has been reported to autonomously generate D-loops[23]. Alignment of RecA and eukaryotic Rad51 sequences shows that the entire C-terminal homology block containing RecA secondary dsDNA binding site residues required for homology probing is not conserved in Rad51[41–43]. We confirm D-loop formation after deproteinization by gel electrophoresis, reaching the highest yield at 60 min with 38%. Interestingly, EM analysis of the DNA-protein complexes at 60 min reveals 58% RecA-mediated joint molecules, with no formation of protein-free D-loops (Fig. 4). This suggests RecA is not displaced from the synapsis during or after the strand invasion and heteroduplex formation, consistent with previous findings[44]. These data are consistent with a specific role of Rad54 in the coordination of the Rad51 displacement along with the transformation of synaptic complexes to D-loops.

EM enables the characterization of the joint molecule architecture as we can precisely determine where the proteins are bound and measure the DNA length as well as the synaptic part of the joint molecules. Analyzing Rad51-mediated synaptic complexes and D-loops we determine the median contact length of synaptic complexes and D-loops (heteroduplex) to be 56.5 nm (mean $67 \pm 6$ nm) and 80 nm (mean $85 \pm 5$ nm), equivalent to 166 bp (or 111 bp when corrected for the extension by Rad51) and 233 bp (B-form DNA), respectively (Fig. 2e). In most of the synaptic complexes, the unpaired ssDNA region after the contact zone is covered with RPA, suggesting Rad51 has dissociated. The topology of the negatively supercoiled dsDNA donor is affected by joint molecule formation as it relaxes as a consequence of synaptic complex and subsequent D-loop formation[2,16,45,46]. The stretching of the donor in the synaptic complex by the Rad51 filament consumes negative supercoiling of the donor duplex. The relaxed Linking Number (Lk) for pUC19 donor plasmid is 276 (mean periodicity 10,5) and the theoretical supercoiling constraint corresponds to −16 with a Lk = 259 (the mean superhelical

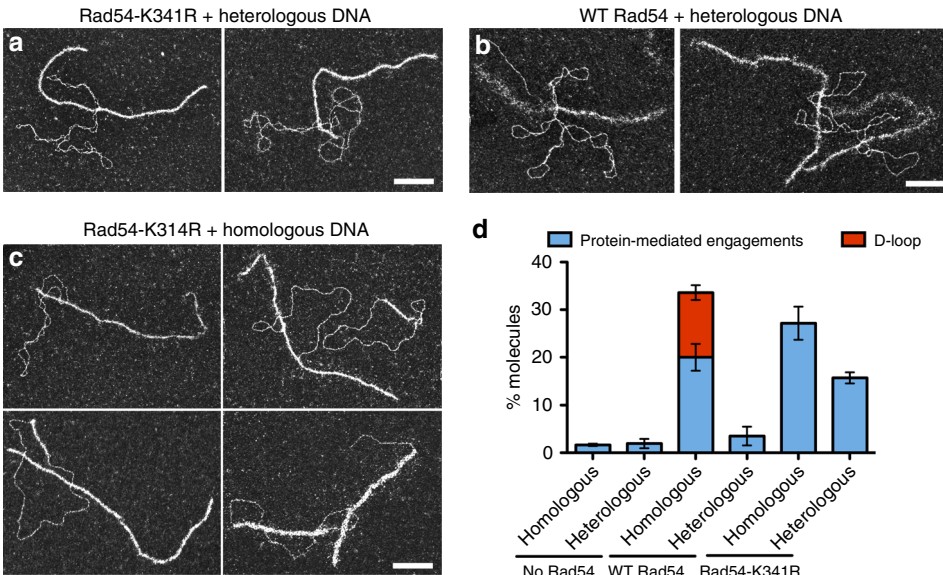

**Fig. 3** Rad54K341R mutant leads to higher yield of protein-mediated engagements with heterologous DNA than WT protein. **a–c** Representative EM pictures of protein-mediated engagements generated during D-loop assays with **a** Rad54K341R and heterologous DNA, **b** wild-type Rad54 and heterologous DNA, and **c** Rad54K341R and homologous DNA. **d** Quantification of joint molecules generated in assays with wild-type Rad54 or Rad54K341R mutant at 20 min ($n = 3$, 250 molecules each $n$). Error bars indicate standard deviation. White bars represent 100 nm. Source data are

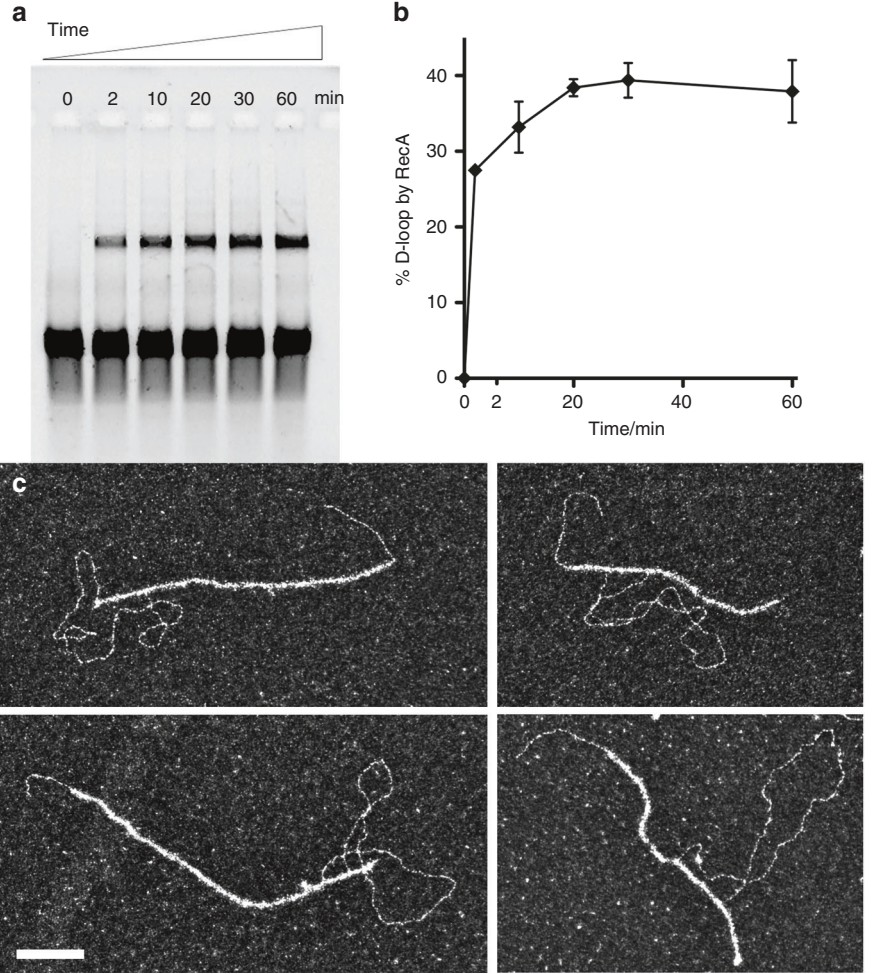

**Fig. 4** RecA promotes heteroduplex DNA formation within the nucleoprotein filament without dissociating. **a** Time course of RecA D-loop assay. **b** Quantification of RecA time time courses; $n = 3$, error bars represent standard deviations. **c** Representative EM pictures of RecA-mediated joint molecules at 20 min. White bar represents 100 nm. Source data are provided as a Source Data file

provided as a Source Data file

density of plasmids extracted from *E. coli* is considered as 0.06[47,48]). This constraint induces a maximal theoretical writhe of 16. In our experiment, the mean writhe (Wr) measured by EM is the number of times DNA double helix crosses itself corresponds to 9. In the synaptic complexes, the absolute value of writhe ranges from 0 to 13. Interestingly, the absolute value of writhe decreases when the contact length increases, showing a significant negative correlation between the synaptic complexes contact length and the writhe (Supplementary Fig. 3). In the DNA-Rad51 filament, the increased length of DNA by 1.5 times and the periodicity of 18.6 bp induces a decrease of Wr to 0.081 for each nm of length filament. In the D-loops, 90% of the molecules present a writhe equal or lower than 2 showing the dissipation of negative supercoils as a consequence of strand intertwining. Interestingly, a subpopulation of synaptic complexes present no change in the whrithe number (Supplementary Fig. 3c) compared to the free plasmid, as if the contact doesn't induce any local double helix opening, suggesting it may involve Rad51-mediated three strands junction without significant effect on the dsDNA homolog topology.

**Rad54-K341R mutant increases heterologous DNA engagements**. To understand how ATP hydrolysis by Rad54 is involved in synaptic complex and D-loop formation, wild-type Rad54 is replaced by its ATP hydrolysis (Walker A box) mutant, Rad54-K341R. This mutant can still bind ATP and stabilize Rad51 filaments but is unable to hydrolyze ATP[31,37]. First, we confirm by gel electrophoretic and EM analysis after deproteinization that D-loop reactions containing this mutant do not yield D-loop products (Fig. 3c, d). Surprisingly, by EM analysis of protein-DNA complexes, this mutant still promotes Rad51-mediated synaptic complex formation with a yield of 27%. Many of these contacts are likely not true synaptic complexes paired via homology since the control heterologous reaction contains ~16% pairings with this mutant Rad54. However, the 12% enrichment in the homology-containing reactions confirms that ATP hydrolysis is not essential for synaptic complex formation by Rad51/Rad54. Similar to reactions lacking Rad54, this mutant allows the formation of short patches of Rad51 filament on the donor dsDNA, consistent with Rad54 removing Rad51 from dsDNA in an ATP-dependent fashion[49].

The enrichment of heterologous pairing in reactions with the Rad54K341R mutant, with 16% protein-mediated pairing (Fig. 3a, d), is surprising. This is four-fold higher than with wild-type Rad54 under identical conditions (4%). Most of these pairings are short and the dsDNA remains supercoiled (high writhe), as expected for heterologous pairing (Fig. 3a). Therefore, the ATPase activity of Rad54 appears to be involved in avoiding persistent heterologous engagements (see below).

**Rad54 is required for Rad51-mediated homology search**. Our analysis of HR protein-mediated DNA pairings by EM reveals Rad54-dependent synaptic complex formation. Very little association with heterologous dsDNA is observed in reactions with wild-type Rad54. However, during homology search, Rad51-ssDNA filaments must engage in very weak and transient interactions with DNA sequences to rapidly find homology among the whole genome. Thus, Rad51 filaments must interact with heterologous sequences to scan them for homology. EM sample preparation includes a dilution step that we suspect induces the loss of weaker, non-homology-mediated interactions between DNA molecules. In order to gain more insight we employed a different approach, the duplex DNA capture assay, to detect protein-mediated interactions of nucleoprotein filaments with dsDNA during homology probing. In reconstituted reactions with

Rad51, RPA, and Rad54, a 5′ biotinylated ds98-ss685 DNA substrate is paired with a supercoiled plasmid DNA and their interaction is assessed by capture of nucleoprotein filaments with streptavidin-coated magnet particles. Associated duplex DNA is quantified by gel electrophoresis and staining (Fig. 5a). Surprisingly, Rad51-ssDNA filaments alone do not associate with duplex DNA as measured by this assay (Fig. 5b, 0 Rad54 data point). In contrast, RecA nucleoprotein filaments capture dsDNA in the presence of either ATP or its slowly-hydrolyzable analog ATPγS (Fig. 5c). This positive control demonstrates that a DNA strand-exchange protein-ssDNA filament can lead to detectable dsDNA binding in this assay. Tellingly, when Rad54 is included in the reaction with Rad51 filaments, the nucleoprotein filaments are found associated with duplex DNA, peaking at ~30% capture at ~50 nM Rad54, substoichimetric to Rad51, present at 260 nM (Fig. 5b). Rad54K341R also supports duplex capture, although it is about two-fold reduced in this activity (Fig. 5b). Duplex capture requires the species-specific Rad51:Rad54 interaction, as human RAD54 does not support duplex capture by yeast Rad51 filaments (Fig. 5b). Control reactions (Fig. 5d) demonstrate that the duplex capture is not significantly affected by the presence of homology in the plasmid, except that homology enables some D-loops to be produced by wild-type Rad54 (3 ± 1%). The complete dependence of duplex capture on both Rad54 (Fig. 5b) and Rad51 (Fig. 5d) rules out that Rad54 is simply binding simultaneously to both ssDNA and dsDNA outside the context of the Rad51-ssDNA filament. In a time course experiment, no time-dependence to the capture signal is observed (Fig. 5e). These finding are consistent with a duplex DNA probing mechanism by the Rad51/Rad54 nucleoprotein complex, presumably in search for homology.

## Discussion

EM and biochemical analysis in this study uncover that yeast Rad54 works along with Rad51-ssDNA filaments in the homology search and synaptic complex formation steps of HR. These functions are in addition to its established ATPase-dependent role in D-loop formation. We propose a more comprehensive model for the cooperative actions of Rad51/Rad54 that culminate in the generation of the critical D-loop intermediate (Fig. 6). (i) During homology search, Rad54 promotes DNA probing. It was an unanticipated finding that yeast Rad51-ssDNA filaments do not interact autonomously with dsDNA but do so robustly in the presence of Rad54 (Fig. 5b). RecA was fully capable to capture duplex DNA using the very same assay (Fig. 5c). This interaction is not dependent on homology (Fig. 5d). We propose this is representative of a probing interaction during homology search, where Rad54 acts as bridging factor of Rad51 to dsDNA, providing a function analogous to the secondary duplex DNA binding site of RecA[35,39,41]. Rad54 ATPase activity may enhance duplex interaction (Fig. 5b), though it is not required. (ii) The Rad54 ATPase avoids persistent contacts of the nucleoprotein filament with heterologous DNA. Homology search involves sampling of heterologous genomic DNA. These contacts need to be transient in order to not slow down the homology search. By EM, we discover that the Rad54K341R mutant leads to higher amounts of persistent heterologous associations. The ATPase activity may enhance the release of heterologous DNA during homology search and/or prevent such stable heterologous associations from occurring (Fig. 3). Interestingly, the situation is reversed in the duplex capture assay with the mutant Rad54 supporting less detectable association of Rad51 filaments with dsDNA compared to wild-type (Fig. 5). This suggests that the mutant Rad54 has less capacity to support duplex DNA probing by Rad51 filaments but the heterologous DNA

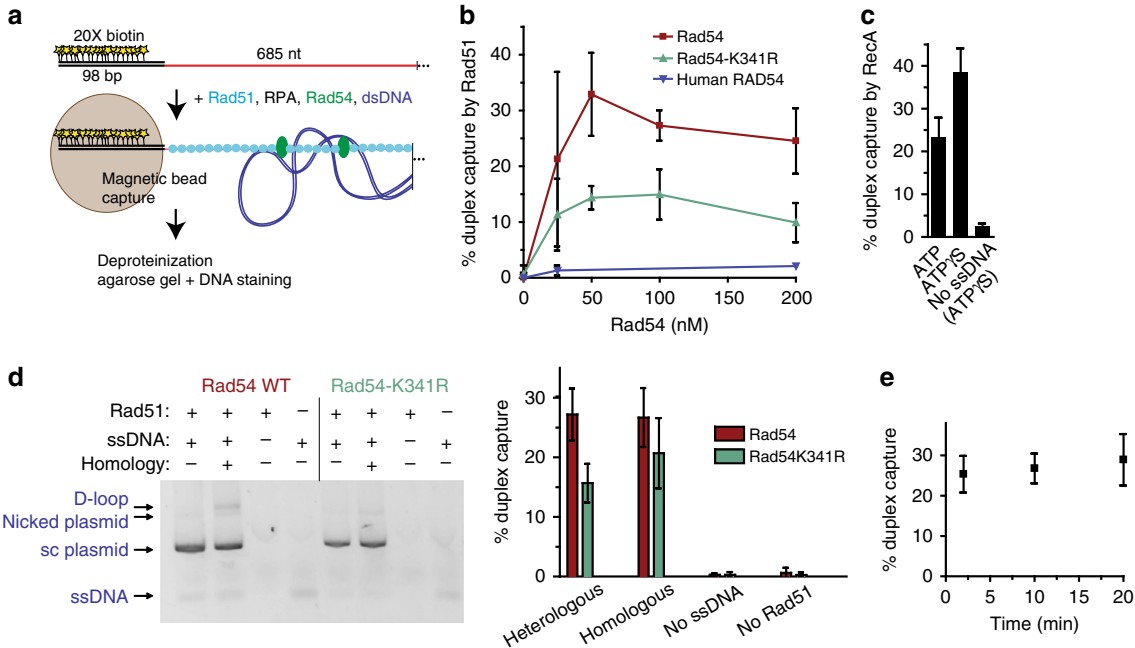

**Fig. 5** Duplex capture by Rad51 filaments is dependent on Rad54. **a** Duplex capture assay. Proteins were incubated at 30 °C, and the order of addition was 10 min Rad51 (260 nM), 9 min RPA (27 nM), 1 min Rad54 (100 nM or as indicated), 10 min 3 kb supercoiled plasmid DNA (1 nM molecules, heterologous unless noted), 1 min streptavidin-coated magnetic particles before magnetic capture, followed by deproteinization of the isolated bead-bound material for gel electrophoresis/staining to measure captured plasmid DNA. **b** Titration of Rad54 in the duplex capture reaction. **c** RecA supported duplex capture. 1 mM ATP or ATPγS was included in reactions. **d** Control reactions performed with 100 nM Rad54 wild-type or Rad54K341R protein. Homologous and heterologous plasmids are both ~3 kb. The homologous plasmid has 607 nt of homology to the ds98-607-ss78 (=ds98-ss685). D-loops migrate slightly slower than nicked plasmid, as previously established. Representative gel image (inverted signal) is shown, with quantitation of replicate experiments. **e** Time course of duplex capture by Rad51 with 100 nM Rad54. Beads were added one minute prior to the indicated time points, at which they were captured. Data in **b**–**e** are means ± standard deviation of three or more independent experiments. Source data are provided as a Source Data file

associations it promotes are more stable than in the presence of wild-type Rad54. This suggests that Rad54 exerts quality control during homology search. (iii) Rad54 is critical for synaptic complex assembly. EM analysis of protein-DNA complexes revealed that Rad51-mediated synaptic complexes between homologously-aligned DNA requires Rad54 (Fig. 2). Like DNA probing, the Rad54 ATPase activity is not essential in this function. Thus, Rad54's role in synaptic complex assembly might simply reflect the requirement of the probing activity; however, a role in maintaining the integrity of the synaptic complex cannot be ruled out. (iv) Rad54 converts Rad51-mediated synaptic complexes into D-loops using its motor activity. Our EM observations show two main populations of joint molecules in the D-loop assay in the presence of Rad54, characterized by the presence or absence of Rad51 on the synaptic contact, synaptic complexes and D-loops, respectively. This observation directly visualizes a key feature proposed of the model, where Rad54 drives D-loop formation by simultaneously removing Rad51 while generating heteroduplex DNA[21].

It is becoming increasingly clear that yeast Rad51 and Rad54 are inseparable in their HR functions. This was already appreciated in terms of the critical D-loop formation (strand intertwining) step, and this study expands this tight interdependence of activities to the preceding homology probing and synaptic complex formation steps. The similarities of the somatic phenotypes of mutations in *RAD51* and *RAD54* are consistent with a role of Rad54 in homology search and duplex capture, but could also be explained by its critical later role in enabling heteroduplex formation and extension of the invading 3′-end by DNA polymerase. In support of an early role of Rad54 in homology search

and synaptic complex formation are data from in vivo time-resolved ChIP studies showing that absence of Rad54 has little effect on the assembly of Rad51 at the DSB site. However, DSB-distant Rad51 signals, possibly representing homology search intermediates or synaptic complexes at the donor target are strongly reduced in the absence of Rad54 or its homolog Rdh54[39,50]. Another study showed little effect of deleting *RAD54* on Rad51 association with donor site[51], possibly reflecting protocol differences[39,50]. The interpretation of these results is further complicated by the presence of the Rad54 paralog Rdh54/Tid1, with partial overlapping functions[38,50–52]. The fact that *rad54 rdh54* double mutants exhibits stronger phenotypes in the above-mentioned ChIP experiments suggests their cooperation during homology search[39,51].

While phage and bacterial filament proteins are able to form synaptic complexes and D-loops autonomously, eukaryotic Rad51 relies on other proteins to achieve greater complexity of HR regulation. This allows eukaryotes to have both somatic and meiotic recombination pathways that utilize Rad51. In the meiotic HR, the nucleoprotein filament is under special regulation, in part mediated by the meiotic Rad51 paralog Dmc1 along with additional Dmc1-interacting proteins and promotes crossovers with homologous chromosomes instead of sisters[53]. Part of this regulation involves inactivation of the efficient Rad51/Rad54 pair both by Mek1 phosphorylation of Rad54 and competition by Hed1 protein binding to Rad51, blocking the Rad54 interaction[26,53–55]. Our results (Fig. 5) suggest that blocking the Rad51/Rad54 interaction would preclude the ability of Rad51 to participate even in the engagement of genomic DNA in attempt of homology search, freeing the presynaptic filament to allow Dmc1 to be targeted, with help from other proteins, to the homolog.

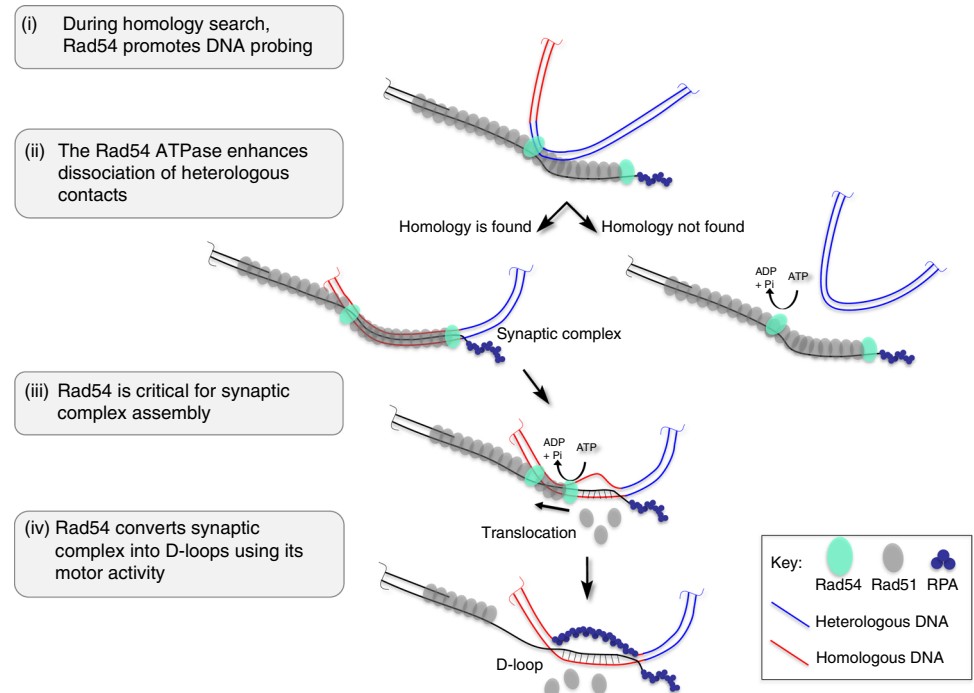

**Fig. 6** Rad51 and Rad54 cooperative model for homology search and D-loop formation. (i) During homology search, Rad54 promotes DNA probing. The invading DNA (light red) uses Rad54 to bridge the Rad51 filament to dsDNA during the homology search. Rad54 ATPase activity is not required but may enhance probing. (ii) Persistent associations with heterologous DNA (blue, right arrow) may be prevented or dissociated by Rad54 in an ATPase-dependent fashion. Rad54 ATPase exerts quality control to promote homologous pairing. (iii) Rad54 is required for synaptic complex formation without strict requirement for ATPase activity, and (iv) converts such complexes into D-loops dependent on ATP hydrolysis. Rad51 left on the ssDNA outside of the heteroduplex region after removal during heteroduplex formation may able to repolymerize back into the synaptic region. Note that this is a cartoon representation not meant to model the true scale and structure of the Rad51 filament or Rad54 protein arrangement in the depicted intermediates

During homology search, contacts of nucleoprotein filaments with heterologous genomic DNA must be transient in order to promote the formation of synaptic complexes. In the case of RecA, the dissociation rate of microhomology (≥8 nt)-mediated DNA pairings is faster in the presence of ATP, suggesting the turnover of non-homologous (or very short homology) duplex DNA by RecA is enhanced by ATP hydrolysis[56]. Yeast Rad51 has a much lower inherent ATPase activity and may rely on other partners to promote heterologous DNA dissociation. Our data suggest that Rad54 and its ATPase activity limit the formation of persistent heterologous associations, as observed by EM (Fig. 3). This does not happen with Rad54K341R mutant, suggesting the suppression of ATPase activity is responsible for this lack of activity. Also differences in the shape/structure of the filaments should be addressed in the future.

In humans, the picture is more complex. In Scanning Force Microscopy (SFM) experiments, human RAD51 filament is able to form joint molecule with homologous dsDNA donor in presence of $Ca^{2+}$ and in the absence of RAD54[57]. No interactions with heterologous donor were observed and it was proposed that interactions between RAD51-coated ssDNA and heterologous dsDNA may be too transient to be detected by SFM. In the joint molecule intermediates, RAD51 remained stably bound to the three strands contact zone supporting the idea that disassembly of RAD51 from the product of recombination requires RAD54. A role of human RAD54 in homology search step still needs to be evaluated. RAD51 has limited capacity to form stable D-loops autonomously, and this reaction requires the presence of $Ca^{2+}$, which inhibits RAD51 ATPase activity stabilizing the filaments[26,58]. It is unclear, whether this is a physiological effect of calcium or whether calcium substitutes RAD51-modulating protein(s) in humans. Rad54 is essential to the yeast reaction, and human RAD51-$Ca^{2+}$ is strongly stimulated by RAD54 in forming D-loops[24,25,27]. As in yeast, human RAD54 could promote D-loop formation by strand intertwining while simultaneously removing Rad51 from the newly formed heteroduplex DNA. However, additional RAD51 partner proteins, such as RAD51AP1, PALB2, and HOP2-MND1, exist that have been reported to promote duplex capture, synaptic complex, and/or D-loop formation by RAD51 filaments[17,18,25,59,60]. Hence, the evidence suggests that the human HR machinery has assumed additional factors, besides RAD54, to achieve D-loop formation.

Single-molecule[6,14] and bulk biochemical experiments[61] have demonstrated that yeast Rad51, as well as bacterial RecA, yeast Dmc1 and human RAD51 can promote homologous pairing in the absence of additional proteins. In these experiments, long Rad51-ssDNAs were preassembled as curtains and their interaction with labeled short (≤70 bp) duplex molecules was visualized with spatial and temporal resolution. Pairing was seen with as little as 8 nt homology[13], which stabilized in a stepping function of triplets when homology was extended[14,16], likely reflective of the non-uniform extension of DNA in the RecA filament in base triplets[2]. It is unclear, whether these intermediates represent strand invasion intermediates and whether the duplex DNA stayed intact. These synaptic complexes are different from the synaptic complexes described here, which are formed between ssDNA-Rad51 filaments only in the presence of Rad54 using a 831 nt ssDNA and a circular 2.4 kbp duplex DNA. We prefer an interpretation that while yeast Rad51 retains some limited capacity to engage dsDNA in some contexts (i.e., has a relatively weak secondary dsDNA binding site), that this is highly enhanced by Rad54, as a bridging factor of the Rad51 filament to dsDNA. A similar argument would apply to the human system of RAD51 interacting proteins discussed above[17,18,25,59,60].

Our EM and biochemical analysis identify a pivotal role of Rad54 in homology search along with the yeast Rad51 protein. This work helps understand the complex interplay of Rad51 filaments with interacting proteins that are vital for HR, which becomes exceedingly complex in humans. This knowledge in turn is crucial for understanding how HR is modulated in cells, which has implications for tumorigenesis, treatment of cancer, and in defining gene editing strategies.

## Methods

**Protein purification.** *Saccharomyces cerevisiae* RPA and Rad51 proteins were purified as described previously[21,62–64]. Yeast Rad54 wild-type and K341R proteins were purified by identical procedures. Thirty-five grams of cells from 10 L yeast culture overexpressing GST-Rad54[64] (pWDH597 = wild-type or pWDH743 = K341R, containing a PreScission protease site between GST and Rad54) from a galactose-inducible promoter were lysed in buffer A containing 20 mM Tris-HCl pH 7.5, 10% glycerol, 1 mM EDTA, 0.5 mM TCEP, and 1 M NaCl with protease inhibitors. Cells were lysed in a bead-beater apparatus (BioSpec) with glass beads (350 g) for eight cycles of 20 s followed by 2 min cooling of the chamber in an ice water bath. Lysate was cleared by centrifugation at 55,000 r.p.m. in a Ti-70 rotor. Cleared lysate (~150 mL) was incubated with 5 mL glutathione resin for 2.5 h. Bound beads were pored into a 1.5 cm diameter column (BioRad) and washed extensively with A/1 M NaCl, then salt was reduced by washing with A/475 mM NaCl. Six milligram of PreScission protease was mixed with the GST-Rad54 bound resin and incubated 1–2 h. Digested material was drained and the first 6 mL developed on an S300 HR column (135 mL) in A/450 mM NaCl. This step removes aggregated protein and DNA that come off in the void volume[65]. Non-aggregated peak fractions were pooled and concentrated to ~1.5 mL in a 15 mL Amicon centrifugal filter device and this material was then dialyzed (and 3X concentrated) in 1 L A/50% glycerol/0.5 M NaCl for 6–8 h. Fifty microliter aliquots were flash frozen in liquid $N_2$ and stored at −80 °C, at a final concentration of 30–50 μM. The total prep time was about 24 h (lysis to freezing), and all steps were carried out at 4 °C.

**Synthesis of 5′ junction DNA construction (609 base pairs with a 3′ overhang of 831 nucleotides).** Two DNA fragments of 1440 and 609 bp were amplified from pBR322 plasmid by PCR using *Taq* polymerase and the pairs of primers Cy5−2574+ × biotin-4014− and biotin-2574+ × 3185−, respectively (Supplementary Table 1). The biotinylated PCR products were purified on a MiniQ 4.6/50 ion exchange column (GE Healthcare Life Sciences) and loaded into a HiTrap Streptavidin HP column (Amersham Biosciences). Purification of the non-biotinylated strand was achieved by elution with 80 mM NaOH, neutralized by addition of HCl 1 M and annealed at equimolar concentrations in molecules, in presence of 1.5 mM $MgCl_2$ then purified on an ion exchange MiniQ column.

**D-loop in vitro assay and analysis of the DNA-protein and DNA intermermediates.** The reaction contained 10 mM Tris-HCl pH7.5, 50 mM NaCl, 3 mM $MgCl_2$, 1.5 mM ATP, 1 mM DTT, 10 mM phosphocreatine, 35 U mL$^{-1}$ phosphocreatine kinase, 25 nM molecules (36 μM nt) of 5′ junction DNA with overhang labeled with Cy5. Rad51 at a final concentration of 12 μM (one protein per 3 nt) was introduced into the reaction and incubated 10 min followed by RPA at a final concentration of 1.44 μM (one protein per 25 nt) also during 10 min. Rad54 (wild-type Rad54 or Rad54K341R as stated in the results) and dsDNA (homologous or heterologous, as stated in the results) were added together in a final concentration of 175 nM and 25 nM in molecules of dsDNA (seven proteins per dsDNA molecule) in a final volume of 14 μL during 20 min. For the homologous donor, pUC19 plasmid was used, while PhiX174 RFI was used as heterologous DNA, both purchased from New England Biolabs and purified on MiniQ ion exchange chromatography column. This assay was carried out at 30 °C. In these reactions lacking Rad54 (control), storage buffer was added in order to rule out non-specific effects of buffer components and to maintain the ionic strength in all samples. For RecA D-loop assay, similar conditions were used, excepted the buffer was replaced by 30 mM of Tris-HCl and 9 mM of $MgCl_2$ and 1 mM of ATPγS with *E. coli* RecA (New England Biolabs) at 12 μM and *E. coli* SSB (New England Biolabs) protein at 0.6 μM.

**D-loop assay analysis by agarose gel electrophoresis.** Seven microliters of the 14 μL reaction were taken and stopped using 0.5 mg mL$^{-1}$ Proteinase K, 1% SDS, 12.5 mM EDTA and incubated overnight at room temperature. A 1% TAE agarose gel was run at 70 V, for 40 min.

**Transmission electron microscopy.** For DNA-protein complexes observation, 2 μL of the remaining 7 μL of reaction were quickly diluted 120 times in a buffer containing 10 mM Tris-HCl pH 7.5, 50 mM NaCl, 3 mM $MgCl_2$ and observed by electron microcopy (DNA-protein samples). For deproteinized reactions suitable for EM observation, the remaining 5 μL were deproteinized using an EDTA and temperature increment treatment based on the addition of EDTA (final

concentration of 50 mM) and its incubation at 40 °C for 10 min followed by 10 min at room temperature. This fraction of the reaction was diluted in the same buffer and analyzed by electron microscopy (so called in the article deproteinized EM samples). In both cases, during 1 min, a 5 μL drop of the dilution was deposited on a 600-mesh copper grid previously covered with a thin carbon film and pre-activated by glow-discharge in the presence of amylamine (Sigma–Aldrich, France)[66,67]. Grids were rinsed and positively stained with aqueous 2% (w/v) uranyl acetate, dried carefully with a filter paper and observed in the annular dark-field mode in zero loss filtered imaging, using a Zeiss 902 transmission electron microscope. Images were captured at a magnification of 85,000× with a Mega-viewIII Veleta CCD camera and analyzed with iTEM software (both Olympus Soft Imaging Solution).

**Rad54 immunolabeling for its EM detection.** To test the presence of Rad54 in the joint molecule structures, we carried out an immunoaffinity labeling procedure. The DNA-protein complexes formed in 7 μL of D-loop reaction were first stabilized using incubation with 0.01% glutaraldehyde cross-linking agent during 10 min at 30 °C. Then 3 μL of a polyclonal anti-yeast Rad54 antibody (primary antibody rabbit IgGs) were added to D-loop reaction and incubated at 25 °C during 10 min followed by 10 min incubation with 5 μM of the secondary immunogold antibody (anti-rabbit). The labeled reaction was then crosslinked with 0.04% glutaraldehyde (0.05% in final concentration) for its subsequent purification by gel filtration on a Smart system (Pharmacia) using a superpose 6 column (Amersham). This last step is essential to remove the excess of proteins and primary and secondary antibodies that are not bound to the complexes. To ensure that the labeling is specific to Rad54 detection, two control experiments were carried out: (1) the same reaction incubated with a control primary polyclonal anti-yeast Srs2 antibody followed with the same procedure, and (2) the reaction in the absence of Rad54. Samples were visualized by EM in dark fields and bright field modes. Note that although this procedure is useful for demonstrating the presence of a protein, the identification remains qualitative.

**Duplex capture assay.** A ds98–607-ss78 DNA substrate with 607 nt homology to phiX174 sequence was produced as described[21], with the following change. For the Klenow 5′ duplex fill-in step, dCTP was replaced with 10 μM biotin-16-dCTP (ChemCyte) to incorporate 20 biotin moieties into the 5′ duplex part of the molecules. These biotinylated substrates were tested for ≥85% capture and liberation using the method described below by analysis of ssDNA recovery on 5% TBE PAGE gels stained with SYBR Gold. Plasmids used below were purified by SDS lysis and CsCl gradient banding.

For capture assays, 1 nM molecules biotinylated ssDNA (0.78 μM nt/bp) was incubated with Rad51 (260 nM) for 10 min at 30 °C in buffer containing 35 mM Tris-HCl pH 7.5, 2 mM ATP, 7 mM $MgCl_2$, 1 mM TCEP, 0.16 mg mL$^{-1}$ BSA, 100 mM NaCl, 20 mM di-tris phosphocreatine and 0.1 mg mL$^{-1}$ phosphocreatine kinase. 27 nM RPA was then added for 9 min, followed by Rad54 (100 nM or as indicated) for 1 min. Next, 1 nM molecules (3 μM bp) of a 3 kb heterologous supercoiled plasmid pBS°−51 (or identical sized homology-containing plasmid, pBS°-phiX) was added for 10 min. To capture ssDNA and any associated duplex DNA, 50 μg (5 μL, resuspended in reaction buffer) of streptavidin-coated magnetic particles (Roche) were added for 1 min, then captured with a magnet. Captured beads were rinsed once with reaction buffer and then resuspended in deproteinization mix/DNA loading dye (2 mg mL$^{-1}$ proteinase K, 0.2% SDS, 10 mM EDTA, 8% glycerol, 1.25% Ficoll 400, ~0.001% each bromophenol blue and xylene cyanol). RecA reactions were performed at 37 °C and contained 260 nM RecA (gift from S. Kowalczykowski, UC Davis), which was allowed to form filaments for 10 min on the ds98-ss685 substrate in buffer containing 35 mM TrisOAc, 10 mM Mg(OAc)$_2$, 0.2 mg mL$^{-1}$ BSA, 1 mM TCEP, ATP regeneration system (10 mM di-tris phosphocreatine and 0.1 mg mL$^{-1}$ phosphocreatine kinase), and either 1 mM ATP or ATPγS. Heterologous plasmid was added for 10 min before capture as above. Human RAD54 was purified as described[27] and was substituted for yeast Rad54 in otherwise identical reactions.

To liberate bound duplex DNA, beads were deproteinized up to 1 h at 30 °C and then overnight at room temperature, heated to 65 °C for 5 min and finally placed directly on ice. This last heat step is not necessary to liberate most bound heterologous plasmid but melts the biotinylated strand of the substrate to allow release of D-loops in homologous donor reactions, as the biotin-streptavidin linkage can persist despite deproteinization treatment. D-loops in supercoiled donors are stable to treatment at 65 °C[21]. Beads were removed with a magnet, and the supernatant containing released plasmid loaded in a 1% TAE agarose gel, along with input standards, and run at 90 V for 60 min and stained with SYBR Gold DNA stain. The% capture was calculated from interpolation from a standard curve fit by second order polynomial in GraphPad Prism software and is reported as percent of input DNA.

**Reporting summary.** Further information on research design is available in the Nature Research Reporting Summary linked to this article.

## Data availability

All relevant data are available from the authors upon reasonable request. The source data underlying Figs. 1, 2, 3, 4 and Supplementary Figs. 1 and 3 are provided as a Source Data file.

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

## Acknowledgements

E.M.T. was supported by an EMBO short-term fellowship and a PhD grant from the French ministry of science and higher education. Work in the W.D.H. laboratory is supported by grants CA92276 and GM58015 from the National Institutes of Health. Work in the E.L.C. laboratory is supported by grants from "la Ligue Nationale Contre le Cancer 78", ANR-13-BSV8-0022 and ANR FIRE ANR-17-CE12-0015.

## Author contributions

Performed the experiments: E.M.T., W.D.W. and P.D. Designed the experiments: E.M.T., W.D.W., W.-D.H., E.L.C. and P.D. Analyzed the data and wrote the manuscript: E.M.T., W.D.W., W.-D.H. and P.D. All authors reviewed and approved the manuscript.

## Additional information

**Competing interests:** The authors declare no competing interests.

