## [Peer Review File · Nature Communications]

Reviewers' comments:

Reviewer #1 (Remarks to the Author):

The novel finding that is described is the identification of so-called synaptic complexes that can only be formed in the presence of Rad54 using a novel duplex capture assay that is capable of detecting less stable protein mediated interactions between rad51-ssDNA and non-homologous dsDNA using primarily EM based analysis. As stated by the authors in line 430 it was an unanticipated finding that yeast Rad51-ssDNA filaments do not interact autonomously with dsDNA but do so robustly in the presence of Rad54. To which extend this relates to the in vivo behavior remains to be established. Interestingly and convincingly shown, the authors identified the accumulation of stable heterologous associations if the ATPase defective variant of Rad54, the Rad54-K341R mutant is used. As such they identify a separation of function for Rad54 using this adapted capture assay, which is to my opinion the most relevant new finding. Given the fact that this synaptic complex formation is a phenomenon that is in this paper described to be found under specific in vitro conditions it remains to be established how it relates to the in vivo interaction where the rad51ssDNA filament has to find and finally exchange information with the correct donor dsDNA.

Specific comments

Line 58: Seems reasonable to use the suggested review as reference here, but addition of a recent independent reference would be recommendable.

Line 301-303: "As expected for a kinetic intermediate preceding the D-loop, synaptic complexes accumulate earlier than D-loop products, which are stable to deproteinization (see below)."

Unclear from the sentence whether synaptic complexes or D-loop products are stable, please rephrase sentence.

Line 317-323: Definition of difference between synaptic complexes and D-loops should be given earlier in the results section. In general the figures are not described in chronological order which makes part of the manuscript difficult to read

Figure 1/2: There seems to be a discrepancy between the frequency of D-loops observed with deproteinization (Figure 1g 20 minutes, 28%) and native conditions (Figure 2d, 14%). Please explain

Figure 4: The control experiments with RecA needs more introduction, since the paper is really focused on yeast Rad51/Rad54 the relevance to the ignorant reader is not clear

Figure 5b: Experiment could be repeated with human RAD54 and human RAD51, possibly human RAD54 does not interact with yeast Rad51?

In EM assays non-hydrolyzable ATP analogs could be used to confirm ATPase activity of RAD54 is indeed essential for D-loop formation

In general, the print quality of the microscopy images is low. High resolution images would be useful to appreciate the fine details of the EM images. Subsequent indication of ssDNA, duplex DNA could also be improved to highlight the strength of these analyses.

In the text figure references are not referred to in the right order (e.g. line 310, 313, 336).

Reviewer #2 (Remarks to the Author):

Dear Authors,

This is a very well-written article with easy to follow flow and language. It is also a nice study in general. I have really enjoyed reading this manuscript.

The experimental details have been carefully thought out and executed, such as not using alkaline lysis-based plasmid purification kits, but SDS lysis and CsCl gradient banding for the purification of the plasmids.

Enough details and references have been included in the methods section that should allow others to reproduce these, or run similar, experiments.

Details included, such as "25 nM in molecules of dsDNA (7 proteins per dsDNA molecule)" make it easier to follow and think about the experiments.

It is great to see that the same experiments were performed using two completely different methodologies (EM and gels), and the results correlate well.

Statistics (percentages supplied from the counts of molecules in the EM images, and replicates) ensure that the EM observations are not random (e.g., substrates had just fallen on each other, which may look like a complex, etc.).

Questions and comments:

- Many of the references to figures and figure panels in the text seem to be mixed-up, and need to be corrected.
- This is probably related to the availability of the equipment and expertise, but I was wondering why the authors used positive staining with UrAc, instead of metal shadowing, which is perfectly suited for such experiments.
- No fields with multiple molecules were shown, but only cut-out single ones. It'd be nice to include a few representative fields in the supplement.
- What is the extent of homology/similarity of the heterologous substrates used as controls, especially when one looks at shorter stretches of the sequences (micro-homology)? May this account for the heterologous substrate some results, though very limited compared to the homologous one?
- Have you tried starting with a linear dsDNA (instead of the 5' DNA junction), and add the 5'->3' exonuclease to convert it to ssDNA? This is what happens in the cells, hence the question (I know it would complicate the experiments, but I am curious to hear whether you have tried this).
- "Rad54 cannot be reliably identified in the structure of the protein:DNA complexes." Wasn't it possible to use a tagged Rad54 and use something like these to label and localize it?
<http://www.nanoprobe.com/products/Ni-NTA-Nanogold-His-tag-labeling-and-detection.html>
- The distribution of the data points in Fig S2c looks bimodal, with a mean around 2 and another one around 10. Have you run a bimodality test on this set? If it really is bimodal, do you have any ideas about why that may be?
- Fig 2c: I really do appreciate the effort in trying to trace and identify different pieces (labeled with blue, red and black), but it should be made clear to the readers that this does not represent an accurate determination, but only a "best-effort" guess, by subjectively evaluating how different parts look like. If different people trace such molecules, they'd probably come up with different outcomes at the junction points, at least in a subset of images.
- Fig 5, order of addition: I am curious to hear whether you have tried adding everything but the DNAs, and add both DNAs at the same time, and if you have, what happened. Alternatively, add everything but the ssDNA, then add the ssDNA. What happens? The latter may mimic what happens in a cell (i.e., physiologically relevant scenario).

- Fig 5e: Capture is already almost saturated at the first time point. Isn't it possible to do shorter times?
- Fig S1 caption has "d) D-loops with visibly displaced strand", but the figure lacks panel(s) 'd(N)'
- Do you imagine a continuous depolymerization of Rad51 by Rad54 from the nucleoprotein filament Rad51 forms, and re-polymerization of Rad51 back to re-form it, making this a very dynamic process?
- I have sent the detailed edits (typos, etc.) to the editor, which should be passed to you in some form.

Best wishes.

We would like to express our gratitude to the reviewers for their helpful comments and positive feedback that led to experiments of add-value and constructive modifications.

Here enclosed, you can find the responses to each reviewer's comments.

Reviewer#1:

The novel finding that is described is the identification of so-called synaptic complexes that can only be formed in the presence of Rad54 using a novel duplex capture assay that is capable of detecting less stable protein mediated interactions between rad51-ssDNA and non-homologous dsDNA using primarily EM based analysis.

As stated by the authors in line 430 it was an unanticipated finding that yeast Rad51-ssDNA filaments do not interact autonomously with dsDNA but do so robustly in the presence of Rad54. To which extend this relates to the *in vivo* behavior remains to be established. Interestingly and convincingly shown, the authors identified the accumulation of stable heterologous associations if the ATPase defective variant of Rad54, the Rad54-K341R mutant is used. As such they identify a separation of function for Rad54 using this adapted capture assay, which is to my opinion the most relevant new finding. Given the fact that this synaptic complex formation is a phenomenon that is in this paper described to be found under specific *in vitro* conditions it remains to be established how it relates to the *in vivo* interaction where the rad51ssDNA filament has to find and finally exchange information with the correct donor dsDNA.

Response:

Addressing the *in vivo* relevance, the most appropriate experiment one could design has been reported by the Jentch lab (Renkawitz J, Lademann CA, Kalocsay M, Jentsch S. Monitoring Homology Search during DNA Double-Strand Break Repair *In Vivo*. Mol Cell. 2013;50: 261–272.), where they perform genome-wide anti-Rad51 ChIP and find that Rad54 and its homolog Rdh54 are required to observe full signal of Rad51 at non-homologous sites throughout the genome. Moreover, it has been shown Rad51 and Rad54 proteins interact *in vivo* and this interaction is important for recombinational repair (Clever et al. 1997. Recombinational repair in yeast: functional interactions between Rad51 and Rad54 proteins).

We changed the text and discussed this more in the discussion.

Specific comments

-Line 58: Seems reasonable to use the suggested review as reference here, but addition of a recent independent reference would be recommendable.

Response: We added a reference (citation 1): Kowalczykowski SC. 2015.

-Line 301-303: “As expected for a kinetic intermediate preceding the D-loop, synaptic complexes accumulate earlier than t D-loop products, which are stable to deproteinization (see below).” Unclear from the sentence whether synaptic complexes or D-loop products are stable, please rephrase sentence.

Response: It has been reworded as: ‘As expected for a kinetic intermediate preceding the D-loop products, synaptic complexes accumulate earlier than D-loops, being these products stable to deproteinization’.

-Line 317-323: Definition of difference between synaptic complexes and D-loops should be given earlier in the results section. In general the figures are not described in chronological order which makes part of the manuscript difficult to read.

Response: In an effort to explain more this difference, we have given the definitions earlier (in the first section of results) and changed the configuration of figure 1. We have also corrected some mistakes concerning the order of figure mention and have clarified the definitions.

-Figure 1/2: There seems to be a discrepancy between the frequency of D-loops observed with deproteinization (Figure 1g 20 minutes, 28%) and native conditions (Figure 2d, 14%). Please explain.

Response: Yes there is, that is why we described three possible explanations in detail: D-loops show a yield of 2% at 2 min, reaching a peak of 14% at 20 min and decreasing to 12% at 30 min (**Fig. 2d**). This D-loop yield is lower than determined by gel electrophoresis and EM of deproteinized samples (**Fig. 1i**). The difference is possibly due to: (i) Rad51 filaments on the synaptic joints being partly disrupted, resulting in an underestimate of D-loops in EM by this measure, (ii) the Rad54 ATPase may allow a subset of formation of heteroduplex within the Rad51 filament (still in a Rad54 ATPase dependent manner), or (iii) Rad51 left on the ssDNA outside of the heteroduplex region after removal during heteroduplex formation is able to repolymerize back into the synaptic region.

-Figure 4: The control experiments with RecA needs more introduction, since the paper is really focused on yeast Rad51/Rad54 the relevance to the ignorant reader is not clear

Response: We have developed this part and the explanation concerning figure 4 and 5 in the 'Results' section.

Precisely, our edition is: 'In order to highlight the specific role of Rad54, we also analyze joint molecules architecture formed during D-loop assay with *E. coli* RecA recombinase, which has been reported to autonomously generate D-loops²³. Alignment of RecA and eukaryotic Rad51 sequences shows that the RecA entire C-terminal block containing secondary dsDNA binding site residues required for homology probing is not conserved in eukaryotic Rad51^{41,42}. We confirm D-loop formation after deproteinization by gel electrophoresis, reaching the highest yield at 60 min with 38 %. Interestingly, EM analysis of the DNA-protein complexes at 60 min reveals 58 % RecA-mediated joint molecules, with no formation of protein-free D-loops (**Fig. 4**). This suggests RecA is not displaced from the synapsis during or after the strand invasion and heteroduplex formation, consistent with previous findings⁴³. **These data are consistent with a specific role of Rad54 in the coordination of the Rad51 displacement along with the transformation of synaptic complexes to D-loops.**'

And in the discussion: 'RecA was fully capable to capture duplex DNA using the very same assay (**Fig. 5c**). This interaction is not dependent on homology (**Fig 5d**). We propose this is representative of a probing interaction during homology search, where Rad54 acts as bridging factor of Rad51 to dsDNA, providing a function analogous to the secondary duplex DNA binding site of RecA^{35,39,41}. Rad54 ATPase activity may enhance duplex interaction (**Fig. 5b**), though it is not required.'

-Figure 5b: Experiment could be repeated with human RAD54 and human RAD51, possibly human RAD54 does not interact with yeast Rad51?

Response: We have preliminary data on this and, rather than stimulate dsDNA capture by hRAD51/calcium, hRAD54 inhibits it. However, the starting signal of RAD51 only is low and requires the presence of calcium or non-hydrolyzable ATP analog (used AMP-PNP). Therefore we have chosen not to include these data in this paper, feeling that this is not a fair comparison to the yeast system. At least we have pointed out in the discussion that likely other factors take on this role in humans based on published data.

-In EM assays non-hydrolyzable ATP analogs could be used to confirm ATPase activity of RAD54 is indeed essential for D-loop formation

Response: We tested the D-loop assay with the ATPgammaS, which is known as a low or non-hydrolysable ATP analog, and we found the D-loop yield is low or none. However, we feel that the Rad54 mutant is already the more appropriate control since the use of ATPgammaS also affects Rad51 dynamics and turnover.

-In general, the print quality of the microscopy images is low. High resolution images would be useful to appreciate the fine details of the EM images. Subsequent indication of ssDNA, duplex DNA could also be improved to highlight the strength of these analyses.

Response: Yes sure, here we submit the original versions of figures with good resolution. We also added arrows to highlight RPA-covered ssDNA and dsDNA as well as Rad51 filament and joint molecules (figure 1).

-In the text figure references are not referred to in the right order (e.g. line 310, 313, 336).

Response: we corrected some mistakes and we reworked the figure order and configuration.

Reviewer #2:

Dear Authors,

This is a very well-written article with easy to follow flow and language. It is also a nice study in general. I have really enjoyed reading this manuscript.

The experimental details have been carefully thought out and executed, such as not using alkaline lysis-based plasmid purification kits, but SDS lysis and CsCl gradient banding for the purification of the plasmids.

Enough details and references have been included in the methods section that should allow others to reproduce these, or run similar, experiments.

Details included, such as “25 nM in molecules of dsDNA (7 proteins per dsDNA molecule)” make it easier to follow and think about the experiments.

It is great to see that the same experiments were performed using two completely different methodologies (EM and gels), and the results correlate well.

Statistics (percentages supplied from the counts of molecules in the EM images, and replicates) ensure that the EM observations are not random (e.g., substrates had just fallen on each other, which may look like a complex, etc.).

Response: Thank you for these positive comments. EM experiments require the quality control of the DNA substrates and proteins and we make a point of honor to detail the methods we use.

Questions and comments:

- Many of the references to figures and figure panels in the text seem to be mixed-up, and need to be corrected.

Response: We have corrected them in this new version.

- This is probably related to the availability of the equipment and expertise, but I was wondering why the authors used positive staining with UrAc, instead of metal shadowing, which is perfectly suited for such experiments.

Response: The application of EM imaging to the study of isolated biological molecules such as protein:DNA complexes has been challenging, mainly because of insufficient contrast and resolution. The use of UrAc staining as well as metal shadowing are contrasting procedures that have usually been applied to the nucleic acids visualization. Annular dark-field combined with UrAc positive staining is the method we have specifically developed to observe DNA for two main reasons: (1) using this staining procedure, DNA molecules are adsorbed onto the carbon film and stained **with no major loss in the tridimensional information**. Uranyl atoms selectively coat the sample allowing an external ultra-structure observation (this is mainly due to a chemical reaction between the uranyl ions and the DNA phosphate groups); (2) this method is practically easier and faster to use.

- No fields with multiple molecules were shown, but only cut-out single ones. It'd be nice to include a few representative fields in the supplement.

Response: We included a field of D-loop reaction final products in figure 1 (panel d).

- What is the extent of homology/similarity of the heterologous substrates used as controls, especially when one looks at shorter stretches of the sequences (micro-homology)? May this

account for the heterologous substrate some results, though very limited compared to the homologous one?

Response: The heterologous substrate we use is the Phix174 plasmid from phage. On the whole 831 nucleotides long ssDNA overhang substrate, we only found one patch of 9 nucleotides and 3 of 8 nucleotides of homology with this plasmid. We cannot totally exclude these microhomologies may account for the heterologous signal (less than 4 % joint molecules). However the traditional D-loop assay has also been carried out with this heterologous plasmid and no D-loop yield was quantified on gel, suggesting that the joint molecules visualized by EM are destabilized upon deproteinization and then are not true D-loop in which complementary strands are aligned and intertwined. We conclude this amount of microhomology is not sufficient for Rad51-Rad54 to promote D-loop formation. But it may be involved in the establishment of rare short contacts between the two molecules.

- Have you tried starting with a linear dsDNA (instead of the 5' DNA junction), and add the 5'→3' exonuclease to convert it to ssDNA? This is what happens in the cells, hence the question (I know it would complicate the experiments, but I am curious to hear whether you have tried this).

Response: We did not perform this type of experiment in this study. To the extent that our goal was to characterize the precise HR intermediate architectures and carry out a statistical analysis, it was important to use a homogeneous DNA substrate that we were also able to follow on D-loop assay gel. The choice of this substrate was mostly based on single strand versus double strand DNA lengths and ssDNA overhang 3' polarity. But we agree the use of exonuclease would be a good manner to progressively generate 3' ssDNA and then to mimic what really happens in the physiological context. In fact we have already performed this type of experiments in other studies (not published yet). According to our expertise, it is difficult to control the exonuclease activity, then the ssDNA length (note that it implies adjusting the buffer to ensure together exonuclease and Rad51 are active) and to avoid that Rad51 dsDNA binding inhibits exonuclease.

- "Rad54 cannot be reliably identified in the structure of the protein:DNA complexes." Wasn't it possible to use a tagged Rad54 and use something like these to label and localize it?

<http://www.nanoprobe.com/products/Ni-NTA-Nanogold-His-tag-labeling-and-detection.html>

Response: Yes, in our new experiments, we succeeded in the specific labeling and identification of Rad54 inside the joint molecules, in the synaptic zone (see Figure 1 and results).

- The distribution of the data points in Fig S2c looks bimodal, with a mean around 2 and another one around 10. Have you run a bimodality test on this set? If it really is bimodal, do you have any ideas about why that may be?

Response: Your question is focused on the new figure S3c. Yes, the distribution of synaptic complexes looks bimodal, more precisely, there is more than one mode. Because there are probably more than 2 modes, we preferred not to apply any statistical bimodality test. But here (see below) you can see the curve we have drawn (using smooth curve on prism), confirming we can group the molecules into 2 main populations: the one where the writhe number doesn't change compared to free plasmid (9) and the one where it decreases. So concerning the effect of strand invasion and synaptic complex formation on dsDNA homolog topology, the most important findings are that (1) there is a subpopulation of synaptic complexes presenting no change in the writhe number compared to the free plasmid, as if the

contact doesn't induce double helix opening, suggesting it may involve Rad51-mediated three strands junction without significant effect on the dsDNA plasmid topology. (2) The more the contact is extended, the more the writhe decreases. The first information wasn't pointed out in the first manuscript we submitted, we have added it in the results section of this revised manuscript.

- Fig 2c: I really do appreciate the effort in trying to trace and identify different pieces (labeled with blue, red and black), but it should be made clear to the readers that this does not represent an accurate determination, but only a “best-effort” guess, by subjectively evaluating how different parts look like. If different people trace such molecules, they'd probably come up with different outcomes at the junction points, at least in a subset of images.

Response: Yes we absolutely agree with you and we added a remark concerning this point in the figure legends. “Note that this division into two categories was performed by EM visual analysis therefore rare subjective analysis-associated errors cannot be rolled out.”

- Fig 5, order of addition: I am curious to hear whether you have tried adding everything but the DNAs, and add both DNAs at the same time, and if you have, what happened. Alternatively, add everything but the ssDNA, then add the ssDNA. What happens? The latter may mimic what happens in a cell (i.e., physiologically relevant scenario).

Response: Both of these orders of addition would inhibit filament growth on ssDNA since Rad51 also avidly binds to duplex DNA (D-loop assay is inhibited in these conditions, EM experiment reveals the formation of short Rad51-dsDNA filament patches). Thus it would require adding in mediator proteins, and since filament formation was not the focus of this study we did not pursue such experiments.

- Fig 5e: Capture is already almost saturated at the first time point. Isn't it possible to do shorter times?

Response: This is definitely true, and in fact this early saturation and lack of time course was the point of the figure panel. We wished to suggest that this signal likely represents dsDNA probing, and hence is not expected to change with time. The nature of the assay, with one minute allowed for the bead capture step does not allow for even shorter timepoints to be reliably taken.

- Fig S1 caption has "d) D-loops with visibly displaced strand", but the figure lacks panel(s) 'd(N)'

Response: 'd(N)' concerns the replicates, not a panel. We clarified the text.

- Do you imagine a continuous depolymerization of Rad51 by Rad54 from the nucleoprotein filament Rad51 forms, and re-polymerization of Rad51 back to re-form it, making this a very dynamic process?

Response: No, we imagine the Rad51 filament as a relatively static scaffold pre-synapsis before incorporation of the invading strand into heteroduplex DNA concomitant with Rad51 removal. Rad54 stabilizes Rad51 on ssDNA, rather than disrupting filaments, reflective also of the fact that it does not have ssDNA-dependent ATPase activity.

- I have sent the detailed edits (typos, etc.) to the editor, which should be passed to you in some form.

Response: Thank you for your attention, we have corrected them.

REVIEWERS' COMMENTS:

Reviewer #1 (Remarks to the Author):

All technical and textual comments from the first review round have been thoroughly and satisfactorily addressed. Furthermore, the abstract has been successfully rewritten to highlight and focus on the main findings. It will be interesting to determine whether the Rad54-K341R ATPase-deficient mutant protein promotes formation of synaptic complexes but not D-loops leading to the accumulation of stable heterologous associations in vivo as well.

Reviewer #2 (Remarks to the Author):

Dear Author(s),

It is with great pleasure that I see my suggestions, such as gold-labeling of Rad54, has contributed to further improving your already well-executed study and well-prepared manuscript.

I have the following minor revision suggestions:

Line 18 - "stabilized on homologous sequence"  "stabilized on a homologous sequence"

Lines 136-137, 139-140 - Statistics given for "contact zone of some joint molecules in reactions with Rad54 (7/100) but none (0/100) in control reactions lacking Rad54 or using a primary antibody directed against another yeast protein (Srs2)." but not for "Rad54 is also specifically detected inside some Rad51 filaments". It'd be nice if this can be added in the same format (i.e., (X/100) after the latter, too).

Figure 6 - As the "homology is found" left to the left arrow, "homology not found" may be added to the right of the right arrow to further clarify the figure.

Best regards.

We would like to thank again the reviewers for their expertise. Here you will find the responses to their second round of comments.

Reviewer #1 (Remarks to the Author):

All technical and textual comments from the first review round have been thoroughly and satisfactorily addressed. Furthermore, the abstract has been successfully rewritten to highlight and focus on the main findings. It will be interesting to determine whether the Rad54-K341R ATPase-deficient mutant protein promotes formation of synaptic complexes but not D-loops leading to the accumulation of stable heterologous associations *in vivo* as well.

Response: We totally agree with your suggestion, it would be really interesting to test this dissociation function mutant *in vivo*. For this purpose, we would first have to develop a new assay to be able to identify and measure synaptic complexes and D-loops formation *in vivo*, which is challenging. To date, we don't know any *in vivo* assay allowing differentiating synaptic complexes from D-loop. We are actually thinking about this. It would be therefore interesting to test Rad54-KR mutant.

Reviewer #2 (Remarks to the Author):

Dear Author(s),

It is with great pleasure that I see my suggestions, such as gold-labeling of Rad54, has contributed to further improving your already well-executed study and well-prepared manuscript.

I have the following minor revision suggestions:

Line 18 - "stabilized on homologous sequence"  "stabilized on a homologous sequence"

Response: thanks we have done the change

Lines 136-137, 139-140 - Statistics given for "contact zone of some joint molecules in reactions with Rad54 (7/100) but none (0/100) in control reactions lacking Rad54 or using a primary antibody directed against another yeast protein (Srs2)." but not for "Rad54 is also specifically detected inside some Rad51 filaments". It'd be nice if this can be added in the same format (i.e., (X/100) after the latter, too).

Response: yes, good suggestion, we have done the quantification and added it to the results

Figure 6 - As the "homology is found" left to the left arrow, "homology not found" may be added to the right of the right arrow to further clarify the figure.

Response: ok

Best regards.